# Efficacy and Safety of Plasma Exchange as an Adjunctive Therapy for Rapidly Progressive IgA Nephropathy and Henoch-Schönlein Purpura Nephritis: A Systematic Review

**DOI:** 10.3390/ijms24043977

**Published:** 2023-02-16

**Authors:** Bryan Nguyen, Chirag Acharya, Supawit Tangpanithandee, Jing Miao, Pajaree Krisanapan, Charat Thongprayoon, Omar Amir, Michael A. Mao, Wisit Cheungpasitporn, Prakrati C. Acharya

**Affiliations:** 1Division of Nephrology, Texas Tech Health Sciences Center El Paso, El Paso, TX 79905, USA; 2Division of Nephrology and Hypertension, Department of Medicine, Mayo Clinic, Rochester, MN 55905, USA; 3Division of Nephrology, Department of Internal Medicine, Thammasat University, Pathum Thani 12120, Thailand; 4Division of Nephrology and Hypertension, Department of Medicine, Mayo Clinic, Jacksonville, FL 32224, USA

**Keywords:** plasmapheresis, apheresis, plasma exchange, IgA nephropathy, vasculitis, Henoch-Schönlein purpura nephritis

## Abstract

Patients with IgA nephropathy (IgAN), including Henoch-Schönlein purpura nephritis (HSP), who present with rapidly progressive glomerulonephritis (RPGN) have a poor prognosis despite aggressive immunosuppressive therapy. The utility of plasmapheresis/plasma exchange (PLEX) for IgAN/HSP is not well established. This systematic review aims to assess the efficacy of PLEX for IgAN and HSP patients with RPGN. A literature search was conducted using MEDLINE, EMBASE, and through Cochrane Database from inception through September 2022. Studies that reported outcomes of PLEX in IgAN or HSP patients with RPGN were enrolled. The protocol for this systematic review is registered with PROSPERO (no. CRD42022356411). The researchers systematically reviewed 38 articles (29 case reports and 9 case series articles) with a total of 102 RPGN patients (64 (62.8%) had IgAN and 38 (37.2%) had HSP). The mean age was 25 years and 69% were males. There was no specific PLEX regimen utilized in these studies, but most patients received at least 3 PLEX sessions that were titrated based on the patient’s response/kidney recovery. The number of PLEX sessions ranged from 3 to 18, and patients additionally received steroids and immunosuppressive treatment (61.6% of patients received cyclophosphamide). Follow-up time ranged from 1 to 120 months, with the majority being followed for at least 2 months after PLEX. Among IgAN patients treated with PLEX, 42.1% (*n* = 27/64) achieved remission; 20.3% (*n* = 13/64) achieved complete remission (CR) and 18.7% (*n* = 12/64) partial remission (PR). 60.9% (*n* = 39/64) progressed to end-stage kidney disease (ESKD). Among HSP patients treated with PLEX, 76.3% (n = 29/38) achieved remission; of these, 68.4% (*n* = 26/38) achieved CR and 7.8% achieved (*n* = 3/38) PR. 23.6% (*n* = 9/38) progressed to ESKD. Among kidney transplant patients, 20% (n = 1/5) achieved remission and 80% (*n* = 4/5) progressed to ESKD. Adjunctive plasmapheresis/plasma exchange with immunosuppressive therapy showed benefits in some HSP patients with RPGN and possible benefits in IgAN patients with RPGN. Future prospective, multi-center, randomized clinical studies are needed to corroborate this systematic review’s findings.

## 1. Introduction

IgA Nephropathy (IgAN), characterized by mesangial accumulation of IgA in kidney biopsy, is the most common type of primary glomerular disease and remains a leading cause of end-stage kidney disease (ESKD) in the world with an estimated incidence of 2.5 per 100,000 persons worldwide [1,2,3,4,5]. The overall prevalence of kidney biopsy-proven IgAN ranges from 4 to 44%, depending on the biopsy criterion and patient descent; the strongest predilection is towards Southeast Asians [1,6,7]. Although synpharyngitic macroscopic hematuria is well recognized as a clinical hallmark of IgAN, the most common initial symptoms in adult patients are microscopic hematuria and/or proteinuria [4,6,8]. The pathophysiology of IgAN is currently considered to be from a multi-“hit” process influenced by genetic and environmental factors [6], resulting in the presence of IgG autoantibodies and galactose-deficient IgA1 circulating immune complexes that deposit in the kidney mesangium. This activates the alternative complement pathway, local inflammation, glomerulosclerosis, and tubulointerstitial fibrosis, resulting in the loss of kidney function [6]. The disease course of IgAN is variable but often slowly progressive; about 25% of cases progress to ESKD within 10 years and about 40% progress within 20 years [9]. The risk of ESKD progression is greater in patients of Southeast Asian descent and those with preexisting risk factors of hypertension, diabetes mellitus, and proteinuria than in patients with different backgrounds [10,11].

IgA vasculitis, also known as Henoch-Schönlein purpura (HSP), is a systemic vasculitis characterized by IgA immune complex deposition within the blood vessels of the affected tissue. HSP is the most prevalent form of vasculitis in children, presenting as rashes, joint pain, gastrointestinal symptoms, and kidney disease. It is usually self-limiting in children but more severe in adults. Kidney biopsy in HSP-associated IgA nephropathy is indistinguishable from that seen in IgAN [4,12,13]. Even though HSP results in greater organ involvement, the risk of ESKD in adults with HSP-associated IgAN is comparable to that of IgAN [14]. 

To date, the 2021 Kidney Disease Improving Global Outcomes (KDIGO) guidelines for glomerulonephritis recommends the use of angiotensin-converting enzyme inhibitors (ACE-i) or angiotensin receptor blockers (ARB) as first-line therapy for the management of all IgAN patients with hypertension or significant proteinuria. Immunosuppressants should be used only if patients remain at high risk for the progression of chronic kidney disease (CKD) despite maximal supportive care or in patients with rapidly progressive glomerulonephritis (RPGN), which is defined as a ≥50% decline in estimated glomerular filtration rate (eGFR) within 3 months. Treatment options to mitigate ESKD progression are still limited for IgAN with crescentic disease [2].

Plasmapheresis or plasma exchange (PLEX) is a therapeutic procedure involving the extracorporeal removal or exchange of blood plasma, which includes its components of antibodies and circulating antigen-antibody complexes [15,16]. PLEX has been beneficial in the treatment of crescentic glomerulonephritis or RPGN due to anti-glomerular basement membrane (GBM) antibody disease and ANCA-associated vasculitis (AAV) [17,18]. Since the pathophysiology of IgAN includes circulating immune complexes, the use of PLEX as adjunctive therapy for IgAN with RPGN could theoretically be advantageous. According to the American Society for Apheresis 2019 guidelines, the role of PLEX may be considered individually in the treatment of IgAN and HSP with rapidly progressive/crescentic (recommendation category III) disease. However, this recommendation is weak due to the lack of randomized/prospective data regarding PLEX use [19]. Thus, this systematic review aims to consolidate existing data and assess the efficacy and safety of PLEX for the treatment of IgAN and HSP-associated IgAN patients with RPGN.

## 2. Materials and Methods

### 2.1. Information Sources and Search Strategy

The protocol for this systematic review is registered with PROSPERO (International Prospective Register of Systematic Reviews; no. CRD42022356411). A systematic literature search was conducted utilizing Ovid Medline, EMBASE, the Cochrane Central Register of Controlled Trials (CCTR), and the Cochrane Database of Systematic Reviews (CDSR) from inception through September 2022 to identify all original studies that investigated the use of PLEX for the treatment of IgAN or HSP with associated RPGN (with or without crescents). Both native and transplanted kidneys affected by IgAN were included. The systematic literature review was individually conducted by two investigators (P.K. and S.T.) using the search strategy as described in the online Supplementary Data. The search strategy included the terms “plasmapheresis or apheresis or plasma exchange” AND “IgA nephropathy or Henoch Schönlein purpura”. A manual search for additional potentially relevant studies using the references of the included articles was also performed. No language limitation was applied. Any differing decisions were resolved by mutual consensus. This study was conducted in agreement with the PRISMA (Preferred Reporting Items for Systematic Reviews and Meta-Analysis) Statement as described in the online Supplementary Data. 

### 2.2. Selection Criteria

Eligible studies included case reports, case series, and cohort studies that evaluated the role of PLEX in the treatment of IgAN or HSP with associated RPGN (with or without crescents). Studies had to report the following outcomes: remissions, relapses, degree of proteinuria, and serum creatinine/estimated glomerular filtration rate. The exclusion criteria included studies that primarily reported other treatment outcomes. Inclusion was not restricted by study size. Remission was determined by the reduction of proteinuria based on each article. In general, complete remission (CR) was defined as proteinuria of less than 0.3 g per 24 h, and partial remission (PR) was defined as a reduction of proteinuria between 0.3 and 3.5 g per 24 h and a 50% reduction from baseline. The quality of each study was evaluated by the investigators using the validated methodological index for non-randomized studies (minors) quality score.

### 2.3. Data Abstraction

A structured data collection report was adopted to derive the following information from the included studies: first author’s name, publication year, country of reporting, demographic data, kidney biopsy features, treatment regimen for PLEX, other treatments given, native or transplanted kidney, the outcome of treatment, adverse effects encountered, and other accompanying disease which would affect the kidneys or would cause alveolar hemorrhage and thrombotic microangiopathy. To ensure precision, this data extraction process was independently performed by three investigators (B.N., P.A, and W.C.)

## 3. Results

After excluding duplications, the search strategy retrieved 1382 potentially relevant articles. After excluding 1275 articles based on the titles and abstracts not fulfilling the inclusion criteria (as described in Figure 1, 107 articles underwent full-length review. An additional 69 articles were excluded due to either a lack of outcome of interest or poor methodological quality. Consequently, 38 studies (29 case reports and 9 case series) with 102 patients were enrolled in the analysis. These 38 studies underwent an assessment of methodological quality utilizing the tool published by Murad et al. in 2018 [20]. The literature retrieval, review, and selection process are shown in Figure 1. The characteristics of all included studies are shown in Table 1 and Table 2. The assessment of methodological quality for each included study is shown in Appendix A.

### 3.1. Effect of Plasmapheresis in Native Kidneys with IgA Nephropathy

Among patients with IgAN, nearly half of the patients (42.1%, *n* = 27/64) achieved remission; of those, 20.3% (*n* = 13/64) achieved CR and 18.7% (*n* = 12/64) achieved PR. The remainder (60.9%, *n* = 39/64) progressed to ESKD.

### 3.2. Effect of Plasmapheresis in Patients with HSP

Among patients with HSP, 76.3% (*n* = 29/38) achieved remission; of those, 68.4% (*n* = 26/38) achieved CR and 7.8% (*n* = 3/38) achieved PR. Only 23.6% (*n* = 9/38) of patients progressed to ESKD.

### 3.3. Effect of Plasmapheresis in Patients with Transplanted Kidneys with IgA Nephropathy

The analysis of the included studies found that only five patients were reported to receive PLEX for IgAN with RPGN in transplanted kidneys. Only one of these five kidney transplant recipients (20%) achieved remission, while the remaining four (80%) developed ESKD.

### 3.4. Alveolar Hemorrhage with IgA Nephropathy

Pulmonary renal syndrome with IgA nephropathy was reported in the included studies. Eleven out of the 84 included patients who were treated with PLEX for IgAN had an alveolar hemorrhage, and the majority of these (10/11) patients had improvement or resolution of pulmonary symptoms after treatment. Four of these patients had a concomitant glomerular disease with IgA nephropathy, including two patients with ANCA positivity, one with Anti GBM antibody positivity, and one with hemolytic uremic syndrome. The role of plasma exchange in pulmonary-renal syndromes for Anti GBM and ANCA vasculitis is well established, and PLEX has been successfully utilized in atypical HUS. Some of the included patients did have concomitant illness along with IgAN (as noted above in Table 1 and Table 2) but overall appeared to have a good response in terms of pulmonary symptoms.

### 3.5. Adverse Events of Plasmapheresis

Infectious complications (8 of 102 patients) were the most commonly reported adverse event. All patients who developed infectious complications were on immunosuppressants, including steroids, mycophenolate, and cyclophosphamide. These infectious complications included catheter-associated sepsis, septic shock, bacterial pneumonia, cytomegalovirus (CMV) viremia, pneumocystis (PJP) pneumonia, influenza A, herpes zoster, and Rothia bacteremia. There were also reports of volume overload and cardiac arrest attributed to hypocalcemia and anaphylaxis. One patient developed an itchy rash following FFP that resolved after treatment with steroids and chlorpheniramine. Another patient developed femoral vein thrombosis, and one patient had PLEX catheter dislodgment. Reported adverse events are summarized in Table 3.

## 4. Discussion

This systematic review demonstrates the potential role of PLEX in the treatment of rapidly progressive/crescentic IgAN and HSP. Plasmapheresis’s removal of immune complexes may have a role in the treatment of aggressive forms of IgA and HSP.

This comprehensive analysis demonstrated a larger benefit with PLEX on HSP compared to IgAN patients (76.3% vs. 42.1% of patients achieved remission, respectively). The underlying reason for this variance is unclear, but it could possibly be related to the underlying pathophysiological differences between these diseases [59]. This study also demonstrated significant heterogeneity in treatment regimens, but a common theme was that early initiation of PLEX was associated with improved renal outcomes. In the Gianviti et al. case series, the five patients who did not achieve remission (reduction of proteinuria to less than 3.5 g per 24 h) did not start PLEX until 2 or more months after the onset of symptoms [40]. In contrast, nine of the ten patients who achieved at least partial remission and had stable kidney function over a follow-up period of 24–72 months had initiated plasmapheresis treatment within 1 month of symptom onset. This relationship was redemonstrated in the Shenoy et al. case series where all patients that initiated PLEX within 2 weeks of symptom onset achieved remission, whereas the single patient who delayed treatment until 2 months after symptom onset ultimately developed ESKD and required a kidney transplant [41]. The findings of this review support the KDIGO 2021 clinical practice guidelines that suggest treating RPGN due to IgA vasculitis similarly to ANCA-associated vasculitis where PLEX is sometimes utilized [2].

IgAN with pulmonary manifestations is rare, but all patients that presented with alveolar hemorrhage in our systematic review had significant improvement or resolution of pulmonary symptoms following PLEX. Although kidney outcomes following PLEX in IgAN with RPGN were ambivalent, these results support the use of PLEX in treating severe extra-renal manifestations of IgAN. PLEX has also been shown to have excellent outcomes in treating extra-renal manifestations of ANCA-associated vasculitis, which highlights the potentially shared pathophysiology between these two diseases [19].

Infectious complications arose in nearly 10% of analyzed patients. While most of the patients were already at high risk of infection due to adjunctive immunosuppression, carefully weighing risks-benefits prior to the initiation of PLEX and monitoring for infection is recommended given the impact of PLEX on both the immune system and antibiotic pharmacokinetics [60].

Crescentic disease was seen in most patients included in this analysis, regardless of IgAN or HSP status. Crescent formation is thought to be related to complement activation; prior studies have highlighted a positive correlation between urinary C4d and the degree of crescent development [61]. A crescent score was added to the Oxford MEST for grading IgAN severity in 2016 after their working group identified an inverse correlation between the degree of crescentic disease and kidney outcome [62]. However, the 2021 KDIGO guidelines recommended that the presence of crescents should not dictate therapy unless there is a concomitant change in eGFR [2]. The analysis failed to demonstrate an association between the degree of crescentic disease and kidney outcomes. Further randomized controlled trials and prospective data are needed to clarify the clinical utility of MEST-C and PLEX.

To the authors’ knowledge, this is the first systematic review of the use of PLEX for rapidly progressive and/or crescentic IgA nephropathy. However, there are several limitations. First, the majority of the published studies are case reports and case series, which often limits data to evaluate long-term outcomes. The literature search for this systematic review did not reveal any published randomized clinical trials that evaluated PLEX in rapidly progressive and/or crescentic GN with IgA nephropathy. Second, the included studies were heterogeneous in terms of the onset of treatment, treatment regimen, patient inclusion, and duration of follow-up. Finally, despite a comprehensive review, only a few kidney transplant patients were included, so the findings of this study may not be generalized for transplant patients.

## 5. Conclusions

In summary, this systematic review supports the benefit of plasmapheresis in HSP with RPGN, and it suggests a possible benefit of plasmapheresis in IgAN with RPGN. Randomized controlled trials are needed to further establish the role of plasmapheresis in rapidly progressive IgA nephropathy.

## Figures and Tables

**Figure 1 ijms-24-03977-f001:**
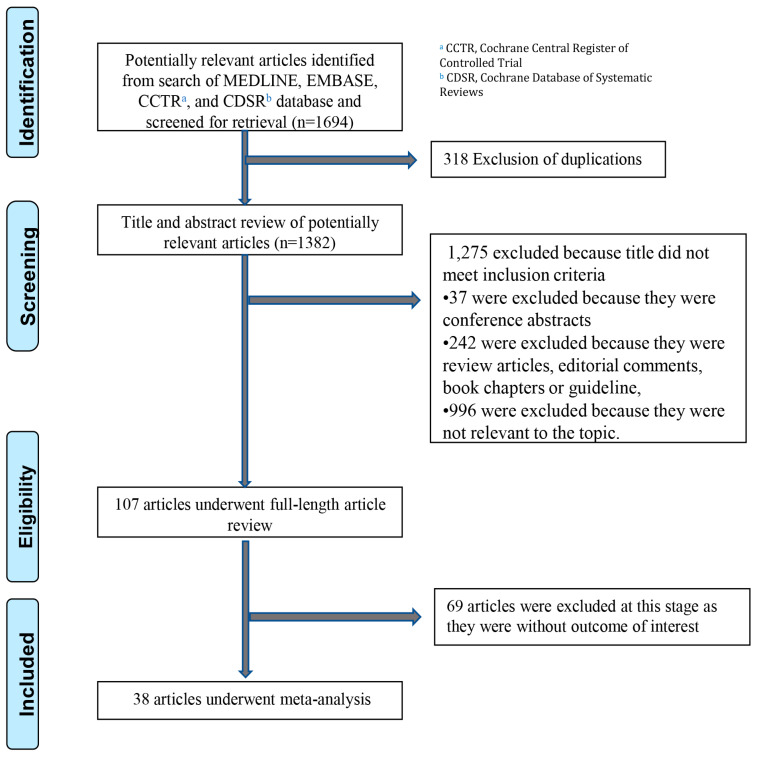
Literature review process.

**Table 1 ijms-24-03977-t001:** Characteristics of included case reports.

	Author	Year	Type of Study	*n*	Country	Age	Sex	HSP	Other Disease	Alveolar Hemorrhage	Crescents	Kidney Transplant	Plasma Exchange Regimen	Additional Treatment	Outcome	Adverse Event	Thrombotic Micro-angiopathy
1	Coppo [21]	1985	Case Report	1	Italy	54	M	-	-	-	20% gloms	-	13 cycles total: 1 session every other day for 3 weeks then weekly sessions for 4 weeks	Steroids Cytoxan	Complete remission Cr clearance improved from 30 mL/min to 120 mL/min Proteinuria 3 g/day to 0.2 g/day at 6-month follow-up	-	-
2	Tejeiro [22]	1990	Case Report	1	Spain	54	M	-	-	-	+60% gloms	+	18 cycles total 22 L removed	Steroids Cytoxan	Not reported	Failed transplant and progressed to ESKD	-
3	Streather [23]	1994	Case Report	1	UK	43	M	-	-	-	+40% gloms	+	3 sessions with 3 L and 4.5% albumin	Steroids	Continued improvement in Cr	-	-
4	Affessa [24]	1997	Case Report	1	USA	66	M	-	-	+	+	-	3× week for 3 weeks	Steroids	Cr 6.9 to 2.8	Catheter dislodged	-
5	McGregor [25]	1998	Case Report	1	New Zealand	14	M	-	P-ANCA (MPO)	+	+90% gloms	-	10 × 2 L exchanges over 3 weeks	Steroids Cytoxan	Cr normal Proteinuria persisted No further pulmonary hemorrhage	-	-
6	Chen [26]	2004	Case Report	1	Taiwan	33	M	+	-	-	+	-	9 sessions of double filtration plasmapheresis	Steroids Cytoxan	S Cr from 11.4 to 3.1	-	-
7	Rech [27]	2005	Case Report	1	Germany	57	M	+	-	-	-	-	3 days first week, 2 days second week, 40 mL/kg with FFP	Steroids Cytoxan	HD until “normal serum creatinine” and resolution of proteinuria at 1 year	-	-
8	Fujinaga [28]	2006	Case Report	1	Japan	5	M	-	-	-	+80% gloms	-	5 sessions alternating days 50 mL/kg	Steroids Mizoribine	HD discontinued 3 weeks after PLEX	-	-
9	Anantham [29]	2007	Case Report	1	Singapore	20	M	-	ESKD due to IgAN	+	+	-	Unclear	Cytoxan Steroids	Improvement in pulmonary hemorrhage, ESKD	-	-
10	Wang [30]	2011	Case Report	1	China	31	F	-	-	-	+14/17 gloms	-	10 sessions	Steroids Cytoxan	Only mentioned Cr 3.75 after 1 mo therapy	-	-
11	Pipilli [31]	2012	Case Report	1	Greece	35	M	-	-	-	+	-	17 sessions	Steroid	Cr from 7 to 2.5	-	+
12	Herzog [32]	2014	Case Report	1	Germany	28	M	-	-	-	+7/12 gloms	-	3 sessions 40 mL/kg	Steroids	ESKD	-	-
13	Otsuka [33]	2014	Case Report	1	Japan	23	M	-	-	-	-	+ 19 days s/p	Double Filtration plasmapheresis	Steroids	Worsening Cr and proteinuria	CMV viremia	-
14	Yim [34]	2014	Case Report	1	Korea	14	M	-	-	+	+21/45 gloms	-	Daily plasmapheresis; weekly for 3 months	PD Steroids Cytoxan	Pulmonary symptoms resolved but progressed to ESKD	-	+
15	Hamilton [35]	2015	Case Report	1	UK	27	M	+	-	-	+20% gloms	-	108 total sessions over 3 years; 2 weeks of daily sessions followed by empiric sessions every 1–2 weeks	Steroids Cytoxan Ritixumab IVIg	Gradual decline in renal function with ESKD at 3 years. Received live renal transplant at 3.5 years with stable Cr of 1.69	-	-
16	Ring [36]	2015	Case Report	1	UK	16	M	+	-	-	+6/14 gloms	-	5 Plasma exchange with 40 mL/kg	Steroids Cytoxan Eculizumab	Not mentioned	No improvement after PLAEX but after Eculizumab, then progressed to ESKD after 2 years	-
17	Doddi [37]	2016	Case Report	1	India	25	F	-	HUS	-	-	-	5 sessions alternate day, 40 mL/kg	-	Cr normal in 3 months	-	+
18	Pannu [38]	2016	Case Report	1	USA	25	M	-	HUS	+	NR	-	PLEX >3 sessions	Eculizumab	Dialysis dependent	-	+
19	Nissaisorakarn [39]	2017	Case Report	1	USA	75	F	-	ANCA	-	+6/13 gloms	-	7 sessions every other day	Steroids Cytoxan	ESKD	Influenza A, Herpes Zoster, Rothia bacteremia	-
20	Soltanpour [40]	2017	Case Report	1	USA	42	M	-	APLS	-	NR	-	PLEX	Steroids	Not reported	Cr improved to 1.9 from 4.5	+
21	Vega [41]	2017	Case Report	1	Spain	69	M	+	-	+	-	-	6	Steroids IVIG 3 mo	Cr 2.1 to 1.2 (unknown time)	-	-
22	Sürmeli-Döven [42]	2018	Case Report	1	Turkey	1.5	M	-	HUS	-	-	-	5 sessions with 1-day intervals	Steroids	Dialysis to Cr 0.52	-	+
23	Rajiv [43]	2018	Case Report	1	India	26	M	-	-	-	+	+	6 sessions	Steroids IVIG Cytoxan	ESKD	-	-
24	Gani [44]	2019	Case Report	1	USA	36	M	-	Humoral and cell-mediated rejection	-	+	+	7 sessions	Steroids Thymoglobulin	ESKD	-	-
25	Kojima [45]	2019	Case Report	1	Japan	66	F	-	Anti GBM	+	+1/18 glom	-	8 sessions	Steroids	ESKD	-	-
26	Longano [46]	2019	Case Report	1	Australia	22	M	-	Anti GBM	+	+2/11 gloms	-	21 sessions	Steroids Cytoxan	Cr remained normal	-	-
27	Bhuwania [47]	2020	Case Report	1	India	58	F	-	ANCA Anti GBM	-	+M1S1C1	-	5 sessions	Steroids Cytoxan (CYCLOPS)	Cr 3.5 to 1.4 at 6 m	-	-
28	Apaydin [48]	2021	Case Report	1	Turkey	18	M	-	COVID PR3ANCA	+	+	-	Daily sessions for 7 days	Steroids IVIg	Cr from 0.96 to 1.15	-	-
29	Zhang [49]	2021	Case Report	1	China	41	F	-	Anti GBM	-	+	-	6 sessions	Steroids Rituximab, HD × 3 IVIG 12 mo Tacrolimus	HD discontinued, Cr 2.79–1.517 at 28 wk	PCP	-

Abbreviations: ANCA, Antineutrophil cytoplasmic antibody; Anti-GBM, Anti glomerular basement membrane disease; APLS, anti-phospholipid disease; Cr, creatinine; ESKD, end-stage kidney disease; Gloms, glomeruli; HD, hemodialysis; HUS, hemolytic uremic syndrome; IgAN, IgA nephropathy; PLEX, plasma exchange therapy.

**Table 2 ijms-24-03977-t002:** Characteristic of included case series.

	Author	Year	Country /Patient no.	Study Population	Age (Yrs)	Other Disease	Initial Kidney Function	Kidney Biopsy	Treatment Regimen	Additional Treatment	Outcome	Adverse Events
1	Lai [50]	1987	UK	2 patients; 2F	21–24	IgAN HTN			Each patient had a different plasma exchange regimen.	Steroids AZA	Both patients saw temporary improvement in serum creatinine following plasma exchange therapy but kidney function gradually deteriorated despite therapy.	Leukopenia
			1	F	24	IgAN HTN	sCr 8.22 mg/dL (727 µmol/L)	20 glomeruli; 13 sclerosed and 7 with fibro-cellular crescents	4 courses consisting of 4 plasma exchanges on alternating days separated by 2–3 months. The first plasma exchange occurred 2 weeks after symptom onset.	Steroids AZA	sCr: 8.14 mg/dL (720 µmol/L) at 3 weeks 4.58 mg/dL (405 µmol/L) at 1 month 9.61 mg/dL (850 µmol/L) at 4 months 5.76 mg/dL (510 µmol/L) at 6 months 10.29 mg/dL (910 µmol/L) at 7 months 5.66 mg/dL (500 µmol/L) at 10 months 11.31 mg/dL (1000 µmol/L) at 12 months ESKD on HD at 15 month follow up	Leukopenia from AZA
			2	F	21	IgAN HTN	sCr 8.22 mg/dL (425 µmol/L)	15 glomeruli; 5 sclerosed 10 with fibro-cellular crescents	6 plasma exchanges on alternating days 2 months after symptom onset.	Steroids AZA	sCr: 5.09 mg/dL (450 µmol/L) at 2 months 9.61 mg/dL (850 µmol/L) at 3 months 5.77 mg/dL (510 µmol/L) at 5 months 5.66 mg/dL (500 µmol/L) at 7 months 6.78 mg/dL (600 µmol/L) at 9 months 7.35 mg/dL (650 µmol/L) at 12 months Progressive deterioration thereafter	None
2	Nicholls [51]	1990	AUS	14 patients; 11M and 3F	17–58	IgAN HTN		All patients had crescents on biopsy with mean of 40% crescents in non-sclerosed glomeruli (median 34%; range 7–80%) No individualized biopsy results were provided	4 plasma exchanges on consecutive days followed by 3 plasma exchanges weekly for 2 weeks, then weekly plasma exchange until 3 months total duration.	Dipyridamole Cytoxan	7 patients experienced fall in sCr during treatment protocol while the renal function of the rest progressively deteriorated during the study. However, all patients ultimately experienced decline in renal function after completion of treatment with all but 4 patients requiring HD. The authors did not provide final outcomes for each individual patient. The 7 patients who had improved with plasma exchange experienced a notably slower rate of decline in renal function compared to the other patients.	Acute Tubular Necrosis in 1 patient
			1	M	18	IgAN HTN	sCr 1.81 mg/dL (160 µmol/L)		Plasma exchange was initiated 3 months after enrollment.		sCr: 2.14 mg/dL (190 µmol/L) at 3 months 2.04 mg/dL (180 µmol/L) at 6 months 2.26 mg/dL (200 µmol/L) at 9 months	
			2	M	23	IgAN HTN	sCr 3.73 mg/dL (330 µmol/L)		Plasma exchange was initiated 3 months after enrollment.		sCr: 4.41 mg/dL (390 µmol/L) at 3 months 4.18 mg/dL (370 µmol/L) at 6 months 4.41 mg/dL (440 µmol/L) at 9 months	
			3	M	30	IgAN HTN	sCr 3.95 mg/dL (350 µmol/L)		Plasma exchange was initiated 3 months after enrollment.		sCr: 6.11 mg/dL (540 µmol/L) at 3 months 5.66 mg/dL (500 µmol/L) at 6 months 7.58 mg/dL (670 µmol/L) at 9 months	
			4	M	26	IgAN HTN	sCr 2.26 mg/dL (200 µmol/L)		Plasma exchange was initiated 3 months after enrollment.		sCr: 2.83 mg/dL (250 µmol/L) at 3 months 1.92 mg/dL (170 µmol/L) at 6 months 2.49 mg/dL (220 µmol/L) at 9 months	
			5	F	40	IgAN HTN	sCr 2.83 mg/dL (250 µmol/L)		Plasma exchange was initiated 3 months after enrollment.		sCr: 10.63 mg/dL (940 µmol/L) at 3 months 7.58 mg/dL (670 µmol/L) at 6 months 20.36 mg/dL (1800 µmol/L) at 9 months	
			6	F	50	IgAN HTN	sCr 1.70 mg/dL (150 µmol/L)		Plasma exchange was initiated 3 months after enrollment.		sCr: 2.37 mg/dL (210 µmol/L) at 3 months 2.03 mg/dL (180 µmol/L) at 6 months 2.26 mg/dL (200 µmol/L) at 9 months	
			7	M	17	IgAN HTN	sCr 6.33 mg/dL (560 µmol/L)		Plasma exchange was initiated 3 months after enrollment.		sCr: 14.37 mg/dL (1270 µmol/L) at 3 months 8.82 mg/dL (780 µmol/L) at 6 months 15.61 mg/dL (1380 µmol/L) at 9 months	
			8	M	58	IgAN HTN	sCr 4.75 mg/dL (420 µmol/L)		Plasma exchange was initiated 3 months after enrollment.		sCr: 5.43 mg/dL (480 µmol/L) at 3 months 5.77 mg/dL (510 µmol/L) at 6 months 7.47 mg/dL (660 µmol/L) at 9 months	
			9	F	20	IgAN HTN	sCr 4.52 mg/dL (400 µmol/L)		Plasma exchange was initiated 3 months after enrollment.		sCr: 5.32 mg/dL (470 µmol/L) at 3 months 8.71 mg/dL (770 µmol/L) at 6 months 12.10 mg/dL (1070 µmol/L) at 9 months	
			10	M	50	IgAN HTN	sCr 2.83 mg/dL (250 µmol/L)		Plasma exchange was initiated 3 months after enrollment.		sCr: 3.28 mg/dL (290 µmol/L) at 3 months 3.28 mg/dL (290 µmol/L) at 6 months 3.39 mg/dL (300 µmol/L) at 9 months	
			11	M	22	IgAN HTN	sCr 4.18 mg/dL (370 µmol/L)		Plasma exchange was initiated 3 months after enrollment.		sCr: 4.18 mg/dL (370 µmol/L) at 3 months 5.43 mg/dL (480 µmol/L) at 6 months 8.03 mg/dL (710 µmol/L) at 9 months	
			12	M	43	IgAN HTN	sCr 7.35 mg/dL (650 µmol/L)		Plasma exchange was initiated 3 months after enrollment.		sCr: 8.03 mg/dL (710 µmol/L) at 3 months 10.29 mg/dL (910 µmol/L) at 6 months 22.51 mg/dL (1990 µmol/L) at 9 months	
			13	M	23	IgAN HTN	sCr 3.96 mg/dL (350 µmol/L)		Plasma exchange was initiated 3 months after enrollment.		sCr: 4.41 mg/dL (390 µmol/L) at 3 months 5.54 mg/dL (490 µmol/L) at 6 months 9.95 mg/dL (880 µmol/L) at 9 months	
			14	M	44	IgAN HTN	sCr 2.37 mg/dL (210 µmol/L)		Plasma exchange was initiated 3 months after enrollment.		sCr: 8.48 mg/dL (750 µmol/L) at 3 months 3.39 mg/dL (300 µmol/L) at 6 months 4.41 mg/dL (390 µmol/L) at 9 months	Developed ATN thought to be related to intercurrent surgery during observation period, but it was withdrawn from analysis.
3	Rocatello [52]	1995	Italy	6 patients; 4M and 2F	16–61	IgAN			All patients except controls in IgAN group received 2 month treatment of 15 mg/kg IV methylprednisolone for 3 days followed by 8 weeks of oral prednisone (1 mg/kg for first 4 weeks and 0.75 mg/kg for last 4) Oral cyclophosphamide 2.5 mg/kg/day for 8 weeks. Plasma exchange (6 treatments in 2 weeks followed by weekly PLEX for at least 2 weeks).	Steroids Cytoxan	All patients saw improvement in serum creatinine and urine abnormalities, but 3 patients eventually developed ESKD at long-term follow up. No correlation between urine abnormalities, HTN, sCr, and histological features was found. No clinical or histological parameter was significantly different between patients in the treatment group.	Pneumonia in 1 patient
			1	M	16	IgAN HTN	sCr 10.0 mg/dL (884 µmol/L)	10 glomeruli 90% florid crescents and 10% fibrotic crescents 1+ interstitial fibrosis	14 plasma exchanges in first month with 8 additional sessions by 2 month follow up.	Steroids Cytoxan	sCr: 2.4 mg/dL (212 µmol/L) at 2 months 2.19 mg/dL (194 µmol/L) at 6 months 5.9 mg/dL (522 µmol/L) at 16 months 7.43 mg/dL (657 µmol/L) at 24 months ESKD on HD at 36 month follow up Repeat biopsy at 16 months: 15 glomeruli 65% glomerular hyalinosis 15% florid crescents 1+ interstitial fibrosis 1+ vascular hyalinosis	-
			2	M	44	IgAN HTN	sCr 1.2 mg/dL (106 µmol/L)	12 glomeruli 15% glomerular hyalinosis 40% florid crescents 1+ interstitial infiltrates 1+ interstitial fibrosis 1+ vascular hyalinosis	11 plasma exchanges in first month, no additional sessions.	Steroids Cytoxan	sCr: 1.1 mg/dL (97 µmol/L) at 2 months 1.49 mg/dL (132 µmol/L) at 6 months 1.49 mg/dL (132 µmol/L) at 24 months Repeat biopsy at 2 months: 26 glomeruli 30% glomerular hyalinosis 10% florid crescents 20% fibrotic crescents 1+ interstitial fibrosis 1+ vascular hyalinosis	-
			3	F	61	IgAN HTN	sCr 7.19 mg/dL (636 µmol/L)	20 glomeruli 5% glomerular hyalinosis 70% florid crescents 1+ interstitial infiltrates 1+ interstitial fibrosis 1+ vascular hyalinosis	14 plasma exchanges in first month, no additional sessions.	Steroids Cytoxan	sCr: 3 mg/dL (265 µmol/L) at 2 months 5.1 mg/dL (451 µmol/L) at 6 months ESKD on HD at 1-year follow up Repeat biopsy at 2 months: 12 glomeruli 30% glomerular hyalinosis 50% florid crescents 1+ interstitial infiltrates 1+ interstitial fibrosis 1+ vascular hyalinosis	-
			4	M	39	IgAN HTN	sCr 2.69 mg/dL (238 µmol/L)	13 glomeruli 35% glomerular hyalinosis 50% florid crescents 1+ Interstitial fibrosis 1+ vascular hyalinosis	10 plasma exchanges in first month with 5 additional sessions by 2 month follow up.	Steroids Cytoxan	sCr: 2.6 mg/dL (230 µmol/L) at 2 months 4.2 mg/dL (371 µmol/L) at 6 months ESKD on HD at 1-year follow up Repeat biopsy at 2 months: 14 glomeruli 30% glomerular hyalinosis 30% florid crescents 2+ interstitial fibrosis 2+ vascular hyalinosis	-
			5	M	55	IgAN HTN	sCr 7.4 mg/dL (654 µmol/L)	10 glomeruli 40% florid crescents 1+ interstitial infiltrates 2+ interstitial fibrosis	10 plasma exchanges in first month, no additional sessions.	Steroids Cytoxan	sCr: 2.19 mg/dL (194 µmol/L) at 2 months 2.09 mg/dL (185 µmol/L) at 6 months 2.19 mg/dL (194 µmol/L) at 24 months 2.19 mg/dL (194 µmol/L) at 36 months ESKD on HD at 1-year follow up No repeat biopsy	-
			6	F	18	IgAN	sCr 3.0 mg/dL (265 µmol/L)	12 glomeruli 15% glomerular hyalinosis 80% florid crescents 1+ interstitial infiltrates 1+ interstitial fibrosis 1+ vascular hyalinosis	18 plasma exchanges in first month with 5 additional sessions between the 2 and 6 months follow up.	Steroids Cytoxan	sCr: 1.49 mg/dL (1.32 µmol/L) at 2 months 2.3 mg/dL (2.03 µmol/L) at 6 months 1.59 mg/dL (1.41 µmol/L) at 24 months 4.2 mg/dL (371 µmol/L) at 120 months No repeat biopsy	-
4	Gianviti [53]	1996	UK	14 patients; 10 M and 4F	3.7–11.9	HSP		12/14 patients: 30–100% crescents	Children weighing below 15 kg underwent plasma filtration with a Gambro plasma filter and AK 10 blood monitor. Children above 15 kg underwent centrifugal plasma exchange with a Cobe Spectra Apheresis system. Total volume exchanged was twice the estimated plasma volume using Albumin and FFP as replacement fluids.	Cytoxan Steroids	All patients with improvement in serum Cr but 5 patients with ESKD at long-term follow up. Statistically significant improvement in kidney outcome if PLEX initiated within 1 month of disease onset.	Volume overload Cardiac arrest due to hypocalcemia Anaphylaxis
			1	F	6.4	HSP	sCr 1.24 mg/dL (110 µmol/L)	60% crescents	9 months from onset	Steroids Cytoxan	sCr 0.53 mg/dL (47 µmol/L) 2 months after PLEX ESKD at 2-year follow up	-
			2	M	9.0	HSP	sCr 2.26 mg/dL (200 µmol/L)	60% crescents	4 months from onset	Steroids Cytoxan	sCr 1 mg/dL (88 µmol/L) 2 months after PLEX ESKD at 2-year follow up	-
			3	M	11.9	HSP	sCr 0.97 mg/dL (86 µmol/L)	80% crescents	1 month from onset	Steroids Cytoxan	sCr 0.68 mg/dL (60 µmol/L) 2 months after PLEX sCr 0.9 mg/dL (80 µmol/L) at 2-year follow up	-
			4	F	9.5	HSP	sCr 5.54 mg/dL (490 µmol/L)	100% crescents	<1 month from onset	Steroids Cytoxan	sCr 1.36 mg/dL (120 µmol/L) 2 months after PLEX sCr 1.92 mg/dL (170 µmol/L) at 6-year follow up	-
			5	M	8.0	HSP	sCr 8.03 mg/dL (710 µmol/L)	80% crescents	1 month from onset	Steroids Cytoxan HD	sCr 1.36 mg/dL (120 µmol/L) 2 months after PLEX sCr (76 µmol/L) at 1-year follow up	-
			6	M	5.1	HSP	sCr 3.73 mg/dL (330 µmol/L)	Diffuse extra-capillary proliferation	<1 month from onset	Steroids Cytoxan HD	sCr (58 µmol/L) 2 months after PLEX sCr 0.86 mg/dL (58 µmol/L) at 2-year follow up	-
			7	M	10	HSP	sCr 8.93 mg/dL (µmol/L)	Diffuse extra-capillary proliferation	1 month from onset	Steroids Cytoxan HD	sCr (62 µmol/L) 2 months after PLEX (53 µmol/L) at 3-year follow up	-
			8	M	8.9	HSP	sCr 1.27 mg/dL (112 µmol/L)	50% crescents	1 month from onset	Steroids Cytoxan	sCr 0.7 mg/dL (41 µmol/L) 2 months after PLEX sCr 0.68 mg/dL (60 µmol/L) at 2-year follow up	-
			9	F	11.5	HSP	sCr 3.39 mg/dL (300 µmol/L)	88% crescents	1 month from onset	Steroids Cytoxan	sCr 0.98 mg/dL (87 µmol/L) 2 months after PLEX sCr 0.66 mg/dL (58 µmol/L) at 2-year follow up	-
			10	M	3.7	HSP	sCr 1.4 mg/dL (124 µmol/L)	30% crescents	48 months from onset	Steroids Cytoxan	sCr 1.36 mg/dL (120 µmol/L) 2 months after PLEX ESKD at 7-year follow up	-
			11	M	5.6	HSP	sCr 2.6 mg/dL (230 µmol/L)	80% crescents	1 month from onset	Steroids Cytoxan	sCr 0.68 mg/dL (60 µmol/L) 2 months after PLEX sCr 0.38 mg/dL (34 µmol/L) at 1.3-year follow up	-
			12	F	10.5	HSP	sCr 2.26 mg/dL (200 µmol/L)	80% crescents	9 months from onset	Steroids Cytoxan	sCr 2.26 mg/dL (200 µmol/L) 2 months after PLEX ESKD at 1-year follow up	-
			13	M	8.5	HSP	sCr 5.32 mg/dL (470 µmol/L)	100% crescents	2 months from onset	Steroids Cytoxan HD	sCr 2.04 mg/dL (180 µmol/L) 2 months after PLEX ESKD at 1-year follow up	-
			14	M	6.7	HSP	sCr 2.6 mg/dL (230 µmol/L)	85% crescents	2 months from onset	Steroids Cytoxan	sCr 0.96 mg/dL (85 µmol/L) 2 months after PLEX 0.97 mg/dL (86 µmol/L) at 9-year follow up	-
5	Shenoy [54]	2007	UK	16 (14 with HSP and 2 IgAN) pts; 6M and 10F	3.7–13.5	HSP IgAN	eGFR estimated using sCr and height		All patients with at least grade 3 nephritis on biopsy were treated with plasmapheresis alone. Plasmapheresis 90 mL/kg per session exchanging 80 mL/kg with 4.5% albumin and 20 mL/kg with FFP. All patients received at least 9 sessions in first 2 weeks with further increasing spaced sessions if clinical recovery was incomplete. All patients received cotrimoxazole 12 mg/kg daily for duration of treatment plus 2 months.	None	All patients had improvement in eGFR and UA/UC ratio that was stable over time, but the delayed patient ultimately required kidney transplant. Results suggest prompt treatment with plasmapheresis alone improves kidney function that remains stable over time.	Itchy rashes following FFP treated with hydrocortisone and chlorphenamine
			1	F	11.0	HSP	eGFR 46	ISKDC grade 3b 20% crescents	Within 2 weeks of onset	None	eGFR 102 with negative urine dipstick for albumin at 7.5 years follow up	-
			2	F	6.8	HSP	eGFR 82	ISKDC grade 3a 40% crescents	Within 2 weeks of onset	None	eGFR 127 and UA/UC 2 at 1.1 year follow up	-
			3	M	5.8	HSP	eGFR 93	ISKDC grade 3b 24% crescents	Within 2 weeks of onset	None	eGFR 98 and UA/UC 3 at 2.1 years follow up	-
			4	M	15.0	HSP	eGFR 20	ISKDC grade 3b 20% crescents	Within 2 weeks of onset	None	eGFR 108 and UA/UC 38 at 2.5 years follow up	-
			5	F	3.7	HSP	eGFR 136	ISKDC grade 3a No crescents	Within 2 weeks of onset	None	eGFR 102 and UA/UC 2 at 6.2 years follow up	-
			6	F	13.5	HSP	eGFR 28	ISKDC grade 4b 53% crescents	Within 2 weeks of onset	None	eGFR 134 and UA/UC 42 at 2.6 years follow up	-
			7	F	12.5	HSP	eGFR 61	ISKDC grade 3b 43% crescents	Within 2 weeks of onset	None	eGFR 101 and UA/UC 10 at 3.1 years follow up	-
			8	M	11.8	HSP	eGFR 33	ISKDC grade 3b no crescents	Within 2 weeks of onset	None	eGFR 142 and UA/UC 1 at 3.8 years follow up	-
			9	M	12.3	HSP	eGFR 90	ISKDC grade 3b 10% crescents	Within 2 weeks of onset	None	eGFR 101 and UA/UC 7 at 1.1 years follow up	-
			10	F	10.1	IgAN	eGFR 42	ISKDC grade 3b 29% fibrous crescents	Within 2 weeks of onset	None	eGFR 106 and UA/UC 2 at 4.2 years follow up	-
			11	M	13.1	IgAN	eGFR 17	ISKDC grade 3b 5% crescents	Within 2 weeks of onset	None	eGFR 113 and UA/UC 16 at 3.4 years follow up	-
			12	M	9.9	HSP	eGFR 43	ISKDC grade 3b 14% fibrous crescents	Within 2 weeks of onset	None	eGFR 105 and UA/UC 9 at 5.2 years follow up	-
			13	F	8.4	HSP	eGFR 64	ISKDC grade 4b 52% crescents	Within 2 weeks of onset	None	eGFR 121 and UA/UC 14.3 at 5.5 years follow up	-
			14	F	8.3	HSP	eGFR 22	ISKDC grade 3a no crescents	Within 2 weeks of onset	None	eGFR 121 and UA/UC 2 at 4.3 years follow up	-
			15	F	8.9	HSP	eGFR 67	ISKDC grade 3b no crescents	Within 2 weeks of onset	None	eGFR 112 and UA/UC 3 at 5.4 years follow up	-
			16	F	7.7	HSP	eGFR 29	ISKDC grade 3b 26% fibrous crescents	Plasma exchange delayed until 2 months from onset due to needle phobia.	None	Kidney Transplant at 6.3 years follow up	-
6	Wright [55]	2006	UK	32 pts; 5 with HSP, gender and specific ages not specified.	Median 9.4 (0.7–17.7 years)	5 with HSP, Rest had collection of PAN, GPA, MPA/ICN, and NCV	eGFR obtained using Schwartz formula		All patients received at least 2 courses of plasma exchange comprised of 5 daily sessions and extra sessions based on clinical response. TPE performed using Spectra centrifugation and PF 1000 plasma filter and Gambro AK 10. Plasma volume was calculated as 50 mL/kg bodyweight with target of double volume as target with limit of 4 L. Plasma replaced with 4.5% albumin in all cases, with FFP at the end of exchange to replenish clotting factors. Median time to treatment from admission was 6 days (range 0–28 days).	Steroids Cytoxan		Hypotension Femoral vein thrombosis Sepsis
			1	Gender not specified	--	HSP	eGFR 64	48% crescents pre-TPE	Did not specify specific time/number of sessions.	Steroids Cytoxan	eGFR 106 after plasma exchange eGFR 162 at 2 months follow up	-
			2	Gender not specified	--	HSP	eGFR 22	100% crescents pre-TPE	Did not specify specific time/number of sessions.	Steroids Cytoxan	eGFR 26 after plasma exchange eGFR 66 at 2 months follow up Required HD temporarily but gradually regained kidney function	-
			3	Gender not specified	--	HSP	eGFR 33	100% crescents pre-TPE	Did not specify specific time/number of sessions.	Steroids Cytoxan	eGFR 20 after plasma exchange eGFR 10 at 2 months follow up Required HD 2 months after plasma exchange	-
			4	Gender not specified	--	HSP	eGFR 167	50% crescents pre-TPE	Did not specify specific time/number of sessions.	Steroids Cytoxan	eGFR 177 after plasma exchange eGFR 169 at 2 months follow up	-
			5	Gender not specified	--	HSP	eGFR 84	75% crescents pre-TPE	Did not specify specific time/number of sessions.	Steroids Cytoxan	eGFR 98 after plasma exchange eGFR 99 at 2 months follow up	-
7	Xie [56]	2016	China	12 patients; 9M and 3F. No individual data available.	Mean 42.7± SD 15	8 patients on HD at start 2 patients with oliguria 11 patients with HTN	Mean sCr 7.98 ± 3.35 mg/dL (705.3 ± 296.4 μmol/L)	Total glomeruli 21 64.4 ± 24.4% crescents; 6 patients 50%< tubular atrophy	Mean 7 sessions (5–10) over mean of 15 days (9–30). 2.517 L exchanged per course (300) Median time of symptoms was 1.5 months (1.0–5.0).	Steroids Cytoxan Some with Mycophenolate	Compared to matched historical control group, about half of plasma exchange group were able to discontinue dialysis in 6 months. 5 patients with significant reduction in sCr to normal range that was stable in long-term follow-up (9 to 51 months). 7 patients with ESKD	Pneumonia Pulmonary Failure
8	Chambers [57]	1999	USA	2 patients								
				M	27	IgAN	2.8 mg/dL (247.58 μmol/L); proteinuria 6.2 g/day	Crescentic GN	6 × 4 L exchanges over 18 days initiated during pt’s readmission.	Steroids, Cytoxan	sCr 5.6 and proteinuria 3.5 g/day, no response to PLEX. ESKD	none
				M	18	IgAN	23 mg/dL (2033.66 μmol/L); >5 g/day	Crescentic GN	7 × 4 L exchanges over 18 days.	Steroids, Cytoxan	ESKD	Sepsis from catheter
9	Rajgopala [58]	2017	India	2 patients								
			1	F	38	DAH	sCr 7.8 mg/dL (689.68 umol/L)	Crescentic GN	Did not specify regimen.	Steroid, Cytoxan	Stable on HD and DAH improved but expired from ventricular arrhythmia during HD on admission day 18	Expired
			2	M	45	DAH	sCr 5.3 mg/dL (689.68 umol/L)	Crescentic GN	Did not specify regimen.	Steroid, Cytoxan, ECMO	DAH not improved; expired from septic shock	Septic shock, Expired

Abbreviations: AZA, Azathioprine; DAH, diffuse alveolar hemorrhage; ECMO, extracorporeal membrane oxygenation; eGFR, estimated glomerular filtration rate; ESKD, end-stage kidney disease; GN, glomerulonephritis; GPA, granulomatosis with polyangiitis; HSP, Henoch-Schönlein purpura; HTN, hypertension; ICN, idiopathic crescentic nephritis; IgAN, IgA nephropathy with Crescentic glomerular involvement; ISKDC, International Study of Kidney Disease in Children; MPA, microscopic polyangiitis; NCV, non-specified vasculitis; PAN, polyarteritis nodosa; PLEX, plasma exchange therapy; sCr, serum creatinine; TPE, therapeutic plasma exchange.

**Table 3 ijms-24-03977-t003:** Reported adverse events.

Adverse Events	Number of Patients
Infectious complication	8 (7.8%)
Mild allergic reaction	1 (0.98%)
Electrolyte abnormality (hypocalcemia)	1 (0.98%)
Catheter dislodgement	1 (0.98%)
Volume overload	1 (0.98%)
Vein thrombosis	1 (0.98%)
Anaphylaxis	1 (0.98%)
Leukopenia	1 (0.98%)

## Data Availability

The data that support the findings of this study are available on request from the corresponding authors.

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
