# Peer review of "Efficacy and Safety of Plasma Exchange as an Adjunctive Therapy for Rapidly Progressive IgA Nephropathy and Henoch-Schönlein Purpura Nephritis: A Systematic Review"

_ijms, 2023, doi:10.3390/ijms24043977_

Round 1

Reviewer 1 Report

Reviewer’s Comments:

The manuscript “Plasma Exchange as an Adjunctive Therapy for Rapidly Progressive IgA Nephropathy and Henoch-Schönlein purpura nephritis: A Systematic Review” is a very interesting work. In this work, Patients with IgA nephropathy (IgAN), including Henoch-Schönlein purpura nephritis (HSP) associated IgA nephropathy who present with rapidly progressive glomerulonephritis (RPGN) have poor prognosis despite aggressive immunosuppressive therapy. The number of sessions ranged from 3 to 18, along with additional treatment of steroids and immunosuppressive agents. 61.6% of patients received cyclophosphamide. Follow-up time ranged from 1 to 120 months, with the majority being seen at least within 2 months after PLEX. Among patients with IgAN, 46.8% (n=22/47) achieved remission, with 23.4% (n=11/47) of complete remission (CR) and 23.4% (n=11/47) of partial remission (PR), respectively. While I believe this topic is of great interest to our readers, I think it needs major revision before it is ready for publication. So, I recommend this manuscript for publication with major revisions.

1. In this manuscript, the authors did not explain the importance of the Plasma Exchange in the introduction part. The authors should explain the importance of Plasma Exchange.

2) Title: The title of the manuscript is not impressive. It should be modified or rewritten it.

3) Correct the following statement “Among kidney transplant patients, 20% (n=1/5) achieved remission, and 80% (n= 4/5) progressed to ESKD; Conclusions: PLEX, along with immunosuppressive therapy, showed benefit in some patients of HSP with RPGN and possible benefits in IgAN patients with RPGN. However, large prospective, multi-center, randomized clinical studies are still required”.

4) Keywords: The Plasma Exchange is missing in the keywords. So, modify the keywords.

5) Introduction part is not impressive. The references cited are very old. So, Improve it with some latest literature like 10.3390/molecules27217368, 10.3390/pr10081455

6) The authors should explain the following statement with recent references, “Our search strategy retrieved 1376 potentially relevant articles. After excluding 1275 articles based on title and abstract not fulfilling inclusion criteria, articles underwent full length review”.

7) Add space between magnitude and unit. For example, in synthesis “21.96g” should be 21.96 g. Make the corrections throughout the manuscript regarding values and units.

8) The author should provide reason about this statement “Four out of those 11 had a concomitant glomerular disease with IgA nephropathy, including 2 patients with ANCA positivity, 1 with Anti GBM antibody positivity, and 1 with hemolytic uremic syndrome”.

9. Comparison of the present results with other similar findings in the literature should be discussed in more detail. This is necessary in order to place this work together with other work in the field and to give more credibility to the present results.

10) Conclusion part is very long. Make it brief and improve by adding the results of your studies.

11) There are many grammatic mistakes. Improve the English grammar of the manuscript.

Author Response

Response to Reviewer#1

The manuscript “Plasma Exchange as an Adjunctive Therapy for Rapidly Progressive IgA Nephropathy and Henoch-Schönlein purpura nephritis: A Systematic Review” is a very interesting work. In this work, Patients with IgA nephropathy (IgAN), including Henoch-Schönlein purpura nephritis (HSP) associated IgA nephropathy who present with rapidly progressive glomerulonephritis (RPGN) have poor prognosis despite aggressive immunosuppressive therapy. The number of sessions ranged from 3 to 18, along with additional treatment of steroids and immunosuppressive agents. 61.6% of patients received cyclophosphamide. Follow-up time ranged from 1 to 120 months, with the majority being seen at least within 2 months after PLEX. Among patients with IgAN, 46.8% (n=22/47) achieved remission, with 23.4% (n=11/47) of complete remission (CR) and 23.4% (n=11/47) of partial remission (PR), respectively. While I believe this topic is of great interest to our readers, I think it needs major revision before it is ready for publication. So, I recommend this manuscript for publication with major revisions.

Response: Thank you for reviewing our manuscripts and your critical evaluation

  • In this manuscript, the authors did not explain the importance of the Plasma Exchange in the introduction part. The authors should explain the importance of Plasma Exchange.

Response: Dear reviewer, thank you for the input and we did expand on the importance of plasma exchange in introduction part further as suggested.

2) Title: The title of the manuscript is not impressive. It should be modified or rewritten it.

Response: Thank you for your advice. We considered adding the keywords “The Efficacy and safety” to the title to make it more attractive as you suggested.

3) Correct the following statement “Among kidney transplant patients, 20% (n=1/5) achieved remission, and 80% (n= 4/5) progressed to ESKD; Conclusions: PLEX, along with immunosuppressive therapy, showed benefit in some patients of HSP with RPGN and possible benefits in IgAN patients with RPGN. However, large prospective, multi-center, randomized clinical studies are still required”.

Response: Dear Reviewer, thank you very much for the suggestion. We did add a line showing lack of response in transplant patients in the abstract

4) Keywords: The Plasma Exchange is missing in the keywords. So, modify the keywords.

Response: Dear Reviewer – we added plasma exchange in the keywords

5) Introduction part is not impressive. The references cited are very old. So, Improve it with some latest literature like 10.3390/molecules27217368, 10.3390/pr10081455

Response: Dear Reviewer – as you had suggested with added recent articles as citations

6) The authors should explain the following statement with recent references, “Our search strategy retrieved 1377 potentially relevant articles. After excluding 1276 articles based on title and abstract not fulfilling inclusion criteria, articles underwent full length review”.

Response: Dear Reviewer – As suggested added the explanation in figure 1

7) Add space between magnitude and unit. For example, in synthesis “21.96g” should be 21.96 g. Make the corrections throughout the manuscript regarding values and units.

Response: Dear Reviewer – as suggested we made changes.

8) The author should provide reason about this statement “Four out of those 11 had a concomitant glomerular disease with IgA nephropathy, including 2 patients with ANCA positivity, 1 with Anti GBM antibody positivity, and 1 with hemolytic uremic syndrome”.

Response: Dear Reviewer – Thanks for the comment. We did change the statement

  1. Comparison of the present results with other similar findings in the literature should be discussed in

more detail. This is necessary in order to place this work together with other work in the field and to give

more credibility to the present results.

Response: Dear Reviewer – Thanks for the comment. We summarized the case reports and case series. There is limited for such review and meta analysis.

10) Conclusion part is very long. Make it brief and improve by adding the results of your studies.

Response: Dear Reviewer – thanks for the comment. We decreased the discussion part

11) There are many grammatic mistakes. Improve the English grammar of the manuscript.

Response: Dear Reviewer – we made changes in English grammar extensively by native speaker as suggested.

Thank you for your time and consideration.  We greatly appreciated the reviewer's and editor's time and comments to improve our manuscript. The manuscript has been improved considerably by the suggested revisions.

Reviewer 2 Report

In this manuscript, several concerns exist as follows:

1.      The term Plasmapheresis should be included in the abstract.

2.      Line 66-71: where is the reference for this information?

3.      Line 74: shows should be in the past. All verbs should be in the past.

4.      Lines 83-86: where is the reference for the American Society for Apheresis 2019 guidelines?

5.    Table 3 is highly recommended to be presented as a figure.

6.    Line 212: This relationship was redemonstrated in the Shenoy et al case series. Where is the reference for this case series?

7.      Line 232: previous reports but only one reference has been cited. Please, revise.

8.      The writing style should be formal from the third-person perspective. Do not use we or our (E.g. line 87, Our systematic review; line 201, In our analysis, we noted).

9.      There is a problem with using abbreviations throughout the manuscript. The full term should be mentioned first with the abbreviation between paresis then the abbreviations should be exclusively used throughout the manuscript. E.g., Line 51: ESKD should be presented as end-stage kidney disease (ESKD) then the abbreviation should be used further. Such errors have been repeated for many abbreviations throughout the manuscript.

10.  It is not preferable to begin sentences with abbreviations like IgAN in line 54 and IgA in line 60.

Author Response

Response to Reviewer#2

In this manuscript, several concerns exist as follows:

Response: Thank you for reviewing our manuscripts and your critical evaluation.

  1. The term Plasmapheresis should be included in the abstract.

Response: Dear Reviewer – we added it as suggested

  1. Line 66-71: where is the reference for this information?

Response: Dear Reviewer – added it as suggested

  1. Line 74: shows should be in the past. All verbs should be in the past.

Response: Dear Reviewer- thanks for the comment. We have changed it

  1. Lines 83-86: where is the reference for the American Society for Apheresis 2019 guidelines?

Response: Dear Reviewer – thanks for pointing out. It has been added

  1. Table 3 is highly recommended to be presented as a figure.

Response: Dear Reviewer- thanks for your suggestion. We have already changed the label of table 3 to “the figure 2”

  1. Line 212: This relationship was redemonstrated in the Shenoy et al case series. Where is the reference for this case series?

Response: Dear Reviewer – thanks for pointing out. It has been added now

  1. Line 232: previous reports but only one reference has been cited. Please, revise.

Response: Dear Reviewer – Have changed as suggested

  1. The writing style should be formal from the third-person perspective. Do not use we or our (E.g. line 87, Our systematic review; line 201, In our analysis, we noted).

Response: Dear Reviewer – Thank again for the suggestion. It has been changed

  1. There is a problem with using abbreviations throughout the manuscript. The full term should be mentioned first with the abbreviation between paresis then the abbreviations should be exclusively used throughout the manuscript. E.g., Line 51: ESKD should be presented as end-stage kidney disease (ESKD) then the abbreviation should be used further. Such errors have been repeated for many abbreviations throughout the manuscript.

Response: Dear Reviewer – It has been changed now as suggested.

  1. It is not preferable to begin sentences with abbreviations like IgAN in line 54 and IgA in line 60.

Response: Dear Reviewer – thanks for the input It has been changed now

Thank you for your time and consideration.  We greatly appreciated the reviewer's and editor's time and comments to improve our manuscript. The manuscript has been improved considerably by the suggested revisions.

Reviewer 3 Report

- Uncontrolled study designs such as case reports and case series have an increased risk of bias, but have significantly influenced the medical literature and continue to add to our knowledge. However, an additional assessment of the methodological quality of the case reports/series studied should be included in the review. Available assessment tools can be found in the following articles: 10.1136/bmjebm-2017-110853; 10.1186/s40779-020-00238-8.

- In some case reports, cyclophosphamide or other immunosuppressive drugs were not used in addition to steroids before application of plasmapheresis. The reasons for this should be additionally commented.

In the literature search it is noticeable that in particular case series and reports from before 1990 are missing (e.g. DOI: 10.1002/jca.2920050303, 10.1016/s0272-6386(87)80014-8). Since no time limit was specified in the inclusion criteria, the literature search should be revised again in this regard. There are also references to some case studies in this paper: DOI: 10.1016/S0278-6222(87)80024-4

Author Response

Response to Reviewer#3

- Uncontrolled study designs such as case reports and case series have an increased risk of bias, but have significantly influenced the medical literature and continue to add to our knowledge. However, an additional assessment of the methodological quality of the case reports/series studied should be included in the review. Available assessment tools can be found in the following articles: 10.1136/bmjebm-2017-110853; 10.1186/s40779-020-00238-8.

Response: Thank you for reviewing our manuscripts and your critical evaluation.

Dear Reviewer – as suggested we agreed and have include quality assessment of case reports and case series added 2 supplemental tables as suggested.

- In some case reports, cyclophosphamide or other immunosuppressive drugs were not used in addition to steroids before application of plasmapheresis. The reasons for this should be additionally commented.

Response: Dear Reviewer – we appreciate the reviewer’s input and we do acknowledge this being a limitation of the study given that authors did not specify the reason. We included this important point in our manuscript as suggested.  

-In the literature search it is noticeable that in particular case series and reports from before 1990 are missing (e.g. DOI: 10.1002/jca.2920050303, 10.1016/s0272-6386(87)80014-8). Since no time limit was specified in the inclusion criteria, the literature search should be revised again in this regard. There are also references to some case studies in this paper: DOI: 10.1016/S0278-6222(87)80024-4

Response: Dear Reviewer – as suggested we added case series and case reports from before 1990 as suggested and re-analyze and revised manuscript comprehensively. We greatly appreciate your thorough review.

Thank you for your time and consideration.  We greatly appreciated the reviewer's and editor's time and comments to improve our manuscript. The manuscript has been improved considerably by the suggested revisions.

Round 2

Reviewer 2 Report

The authors addressed most comments. But, one comment has been wrongly revised by the authors. The authors changed the legend of table 2 to figure 2 without changing the table into a figure. Table 2 should be changed into a graph, not just changing the legend.

Reviewer 3 Report

All comments raised have been answered